

# Shipborne observations reveal contrasting Arctic marine, Arctic terrestrial and Pacific marine aerosol properties

Jiyeon Park[1], Manuel Dall'Osto[2], Kihong Park[3], Yeontae Gim[1], Hyo Jin Kang[1,4], Eunho Jang[1,4], Ki-Tae Park[1], Minsu Park[5], Seong Soo Yum[5], Jinyoung Jung[1], Bang Yong Lee[1], and Young Jun Yoon[1,*]

[1]Korea Polar Research Institute, 26 Songdomirae-ro, Yeonsu-gu, Incheon 21990, South Korea
[2]Institut de Ciències del Mar, CSIC, Pg. Marítim de la Barceloneta 37-49, 08003, Barcelona, Catalonia, Spain
[3]Gwangju Institute of Science and Technology (GIST), 123 Cheomdangwagi-ro, Buk-gu, Gwangju 61005, Republic of Korea
[4]University of Science and Technology (UST), 217 Gajeong-ro, Yuseong-gu, Daejeon, Republic of Korea
[5]Department of Atmospheric Sciences, Yonsei University, 50 Yonsei-ro, Seodaemun-gu, Seoul 03722, Korea

*Correspondence to: Y.J. Yoon (yjyoon@kopri.re.kr)

**Abstract**

There are few shipborne observations addressing the factors influencing the relationships of the formation and growth of aerosol particles with cloud condensation nuclei (CCN) in remote marine environments. In this study, the physical properties of aerosol particles throughout the Arctic Ocean and Pacific Ocean were measured aboard the Korean ice breaker R/V Araon during the summer of 2017 for 25 days. A number of New Particle Formation (NPF) events and growth were frequently observed in both Arctic terrestrial and Arctic marine air masses. By striking contrast, NPF events were not detected in Pacific marine air masses. Three major aerosol categories are therefore discussed: (1) Arctic marine (aerosol number concentration $CN_{2.5}$: 413 ± 442 $cm^{-3}$), (2) Arctic terrestrial ($CN_{2.5}$: 1622 ± 1450 $cm^{-3}$) and (3) Pacific marine ($CN_{2.5}$: 397 ± 185 $cm^{-3}$), following air mass back trajectory analysis. A major conclusion of this study is that not only that the Arctic Ocean is a major source of secondary aerosol formation relative to the Pacific Ocean; but also that open ocean sympagic and terrestrial influenced coastal ecosystems both contribute to shape aerosol size distributions. We suggest that terrestrial ecosystems - including river outflows and tundra - strongly affects aerosol emissions in the Arctic coastal areas, possibly more than anthropogenic Arctic emissions. The increased river discharge, tundra emissions and melting sea ice should be considered in future Arctic atmospheric composition and climate simulations. The average CCN concentrations at a supersaturation ratios of 0.4% were 35 ± 40 $cm^{-3}$, 71 ± 47 $cm^{-3}$, and 204 ± 87 $cm^{-3}$ for Arctic marine, Arctic terrestrial, and Pacific marine aerosol



categories, respectively. Our results aim to help to evaluate how anthropogenic and natural atmospheric
sources and processes affect the aerosol composition and cloud properties.
**1. Introduction**
The climate change experienced in the Arctic is more rapid than that occurring at mid-latitudes in a
phenomenon known as Arctic amplification (ACIA, 2005). In the warming Arctic, the extent and
thickness of sea-ice have dramatically decreased over the past few decades (Stroeve et al., 2012). It has
been estimated that the Arctic may seasonally become sea ice-free Arctic in the next 30 years (Wang
and Overland, 2009). Aerosol particles in the atmosphere are a major driver of the Arctic climate (IPCC,
2013), as they directly affect the climate through scattering and absorbing solar radiation (Stier et al.,
2007), and indirectly by modifying the formation, properties, and lifetimes of clouds (Twomey, 1974).
These direct and indirect effects are the leading uncertainty in current climate predictions. New particle
formation (NPF), a predominant source of atmospheric particles, occurs through the formation of
nanometer-sized molecular clusters (<~1 nm) (i.e., nucleation) and their subsequent growth into aerosol
particles of a few nanometers (~1 – 10 nm) and larger (~>10 nm) (Kulmala et al., 2004;Zhang et al.,
2012). NPF can significantly increase the number of aerosol particles in the atmosphere. During
summer, the Arctic is more isolated from anthropogenic influences (Arctic Haze) and experiences
comparatively pristine background aerosol conditions (Heintzenberg et al., 2015;Law and Stohl, 2007).
As the number concentrations of particles in the Arctic during summer are very low (of an order of ~$10^2$
cm$^{-3}$) (Merikanto et al., 2009), the physicochemical properties of aerosol particles in the Arctic
atmosphere is highly sensitive to NPF.
NPF events have been frequently observed within a wide range of environmental conditions at
various Arctic locations, such as Zeppelin (Tunved et al., 2013;Croft et al., 2016;Heintzenberg et al.,
2017), Tiksi (Asmi et al., 2016), Alert (Croft et al., 2016), Station Nord (Nguyen et al., 2016), and
Barrow (Kolesar et al., 2017), and from limited ship-based observations (Chang et al., 2011;Kim et al.,
2015;Heintzenberg et al., 2015). The formation and growth of particles in the Arctic atmosphere are





strongly influenced by marine, coastal, marginal ice, and/or anthropogenic sources. Oceanic dimethyl
sulfide (DMS) and other volatile organic precursors (such as isoprene, monoterpenes, and amines) play
important roles in the formation and growth of new particles in the Arctic (Leaitch et al., 2013;Willis et
al., 2016;Park et al., 2017;Abbatt et al., 2019;Mungall et al., 2016). In addition, iodine oxides
significantly contribute to NPF in marine and coastal Arctic environments owing to emissions from
marine microalgae at low tide or snowpack photochemistry in ice-covered regions (Allan et al.,
2015;O'Dowd et al., 2002;Raso et al., 2017). Biogenic gaseous precursors released by the melting
Arctic sea-ice margins have also been associated with NPF (Dall´Osto et al., 2017;Willis et al., 2018).
Recent studies in Alaska have indicated that the formation and growth of particles are influenced by
emissions from oil and gas extraction activities in Prudhoe Bay (Gunsch et al., 2017;Kolesar et al.,
2017). Although several observations have been made in the Arctic under different environmental
conditions, there are few detailed size distribution analyses of particle formation and growth events
within the Arctic marine environment.

Several studies have attempted to investigate the impacts of NPF on the concentrations of cloud

condensation nuclei (CCN) (Willis et al., 2016;Rose et al., 2017). Model-based studies have predicted
that a large fraction of CCN (up to 78% of CCN at 0.2 % supersaturation) in the global atmosphere
results from atmospheric NPF and growth (Merikanto et al., 2009;Westervelt et al., 2014;Spracklen et
al., 2008). Field observations have also observed substantial increases in the concentrations of CCN due
to atmospheric nucleation in various environments (Pierce et al., 2012;Kalivitis et al., 2015;Kim et al.,
2019). Several examples of increases in the CCN concentrations after a few hours from the beginning of
NPF events were presented by Kim et al (2019) at King Sejong Station in Antarctic Peninsula, by Pierce
et al. (2012) in a forested mountain valley in western Canada, by Kalivitis et al. (2015) at an eastern
Mediterranean atmosphere in Grete, Greece, by Willis et al. (2016) in Arctic aircraft campaign in
Nunavut, Canada, and by Rose et al. (2017) at the highest atmospheric observatory on Chacaltaya,
Bolivia. However, due to the infrequency of aerosol measurements collected onboard ice breakers, very
few studies have measured the simultaneous aerosol size distribution and CCN concentrations over the





Arctic Ocean.
In this study, the physical characteristics of aerosol particles over the Arctic and Pacific Oceans were
investigated between August 26 and September 24, 2017, using aerosol particle monitoring instruments
installed on the Korean ice breaker R/V Araon. Data of the aerosol size distribution, the concentrations
of the total aerosol number (CN), black carbon (BC), and CCN were continuously collected using
various aerosol instruments. The main aims of this study were to (1) investigate the frequency and
characteristics of NPF and particle growth over the Arctic and Pacific Oceans, (2) determine the major
sources that are associated with NPF based on backward air mass trajectory analysis, and (3) explore the
potential contribution of NPF to the CCN concentrations in the remote marine environment.
**2. Experimental methods**
**2.1. Study area and ship tracks**
Ambient atmospheric aerosol measurements were collected over the Arctic and Pacific Oceans
onboard the ice breaker R/V Araon, operated by the Korea Polar Research Institute (KOPRI), Korea.
The ship's track is presented in Fig. 1. The cruises covered two main areas: the Arctic Ocean (including
both Beaufort and Chukchi Seas) and the remote Northwest Pacific Ocean. The ship departed from
Barrow, USA, on August 28, 2017, crossed the Beaufort (August 29-September 13, 2017) and Chukchi
Seas (September 15, 2017), and reached Nome, USA, on September 16, 2017. The Beaufort Sea
extends across the northern coasts of Alaska and the Northwest Territories of Canada. After completing
the Arctic survey, the ship departed from Nome, USA, on September 18, 2017, crossed the Bering Sea,
Sea of Okhotsk, and East Sea, and reached Busan, Korea, on September 28, 2017.
**2.2. Atmospheric aerosol measurements**
The aerosol sampling inlet was placed on the front deck of the ship (13 m above sea level), ahead of
the ship's engines to avoid any influences from the emissions of the ship's exhaust. In addition, kitchen
ventilation systems were connected by a plastic cylindrical pipe (~15 m length) and moved back on the



deck (far away from the sampling inlet) to minimize the potential effects of cooking emissions on the
atmospheric measurements during the sampling periods. Aerosols were sampled through a stainless
steel tube (inner diameter of 1/4 in, and length of ~1 m), which was connected to the various
instruments by electrically conductive tubing to minimize particle losses in the sampling line.
The physical properties of the aerosols were measured with various aerosol instruments, including
two condensation particle counters (TSI 3776 CPC and TSI 3772 CPC), two scanning mobility particle
sizers (SMPS), an optical particle sizer (OPS), an aethalometer, and a cloud condensation nuclei counter
(CCNC). The TSI 3776 CPC and TSI 3772 CPC measured the total number concentrations of particles
larger than 2.5 and 10 nm every 1 sec, respectively. The aerosol sample flow rates of TSI 3776 CPC and
TSI 3776 CPC were 1.5 and 1.0 lpm, respectively. The number size distributions of the particles were
measured using the nano SMPS every 3 min (Differential mobility analyzer (DMA): TSI 3085, CPC:
TSI 3776), covering a size range of 3 to 80 nm, and the standard SMPS (DMA: TSI 3081, CPC: TSI
3772) every 3 min, covering a size range of 10 to 300 nm. The aerosol and sheath flow rates of the
nano-SMPS were 1.5 and 15 lpm, respectively; and those of the standard SMPS were 1.0 and 10 lpm,
respectively. An OPS (TSI 3330) was used to determine the size distribution of particles in a range of
100 nm to 10 μm with a sample flow rate of 1.0 lpm. The BC concentration was measured using an
aethalometer (AE22, Magee Scientific Co., USA) to assess the influence of anthropogenic sources (such
as local pollution and ship emissions). The instrument uses the absorption of light at a wavelength of
880 nm by the ambient aerosols collected on a quartz filter tape to determine the BC concentration. The
flow rate through a sharp-cut 2.5 μm cyclone (BGI, Inc., USA) was set to 5 lpm and the integration time
was 5 min. The Droplet Measurement Technologies CCN counter (DMT CCN-100) was operated to
measure the CCN number concentrations. The total flow rate in the CCN counter was 0.5 lpm, and the
counter was operated at five different supersaturation ratios (SS) (0.2, 0.4, 0.6, 0.8, and 1.0 %) every 30
min. The sample and sheath flow rates of the CCN counter were 0.05 and 0.45 lpm, respectively.

**2.3. Identification of ship exhaust**





To obtain a data set that reflects background aerosol loading, measurement data affected by the
exhaust emissions of the ship's engine should be excluded prior to further data analysis. For this,
aerosol data were filtered based on the BC concentration, wind direction, wind speed, and total particle
number concentration. The data with the following properties were discarded: (1) BC concentrations
exceeding 100 ng m$^{-3}$, (2) relative wind direction against the ship's heading between 110° and 260°, as
this originates directly from the ship's exhaust, (3) relative wind speed lower than 2 m sec$^{-1}$ as air
masses under a calm environment could become contaminated due to local turbulence, and (4) the total
particle number concentrations were particularly high (spike) and varied dramatically in a short time.
Ship plumes were clearly observed in the data collected during the campaign. Typically, the ship
exhaust differs from the NPF events as the enhanced number concentration during the NPF events
lasted for at least an hour with a low BC concentration (Ehn et al., 2010).

**2.4. Backward air mass trajectory and satellite observations**
The backward air mass trajectories were analyzed using version 4 of the Hybrid Single-Particle
Lagrangian Integrated Trajectory (HYSPLIT) model (http://ready. arl.noaa.gov/) to examine their
relationships with the physical characteristics of aerosol particles. The 2-day air mass back trajectories
(48 h) were determined at hourly intervals from the ship's position at an arrival height of 50 m to
estimate the transport history of the air masses arriving at the observation site (Park et al., 2018). The
potential origins of the aerosols were divided into three categories based on the retention time of the 2-
day back trajectories over the three major domains: Arctic Ocean (including the Beaufort and Chukchi
Seas, and sea-ice region), Pacific Ocean (including the Bering Sea and Sea of Okhotsk) and land
(including Alaska and the eastern part of Siberia) (Fig. 1). The phytoplankton biomass was obtained by
calculating the chlorophyll-*a* concentration from the level-3 product of Aqua Moderate Resolution
Imaging Spectroradiometer at a 4 km resolution (Fig. S1). Geographical information over the ocean,
land and sea-ice was obtained from the sea-ice index, which was provided by the National snow and Ice
Data Center (NSIDC) (Fig. S2). Note that the sea-ice extent was defined as the area having an ice





concentration of ≥ 15% (Pang et al., 2018). Air masses that intensively passed over the Beaufort and
Chukchi Seas and sea-ice region were categorized as Arctic Ocean originated air masses (i.e., > 50%
retention over the ocean > 65°N and sea-ice region). Air masses that intensively passed over Northern
Alaska and the eastern Siberia were potentially affected by the Arctic tundra and categorized as land
originated air masses (i.e., > 50% retention over the land domain). Finally, air masses that traveled
through the Bering Sea and Sea of Okhotsk were categorized as air masses originated from Pacific
Ocean domain (i.e., > 50% retention over the ocean domain < 65°N).
**2.5. Oceanic measurements**
To examine the influence of oceanic conditions on NPF and growth, seawater samples were collected
from sea surface at a depth of ~ 1 m by Niskin bottles. The sampling locations and methods have been
described previously in more detailed (Park et al., 2019). In brief, concentrations of dissolved organic
carbon (DOC) were measured with a Shimadzu TOC-V high-temperature combustion total organic
carbon analyzer. To identify the source and composition of DOC in surface seawater, three-dimensional
excitation-emission matrixes (EEMs) were scanned using a fluorescence spectrometer (Varian, USA).
The excitation wavelength range was between 250 and 500 nm, and emission between 280 and 600 nm.
In this study, the four major fluorescent components were classified into 4 groups; terrestrial humic
substances peak (A) (EX: 260 nm, EM: 380−460 nm), the terrestrial fulvic substances peak (C) (EX:
350 nm, EM: 420−480 nm), the marine fulvic substances peak (M) (EX: 312 nm, EM: 380−420 nm),
and the proteinaceous peak (T) (EX: 275 nm, EM: 340 nm) (Coble, 2007).
**3. Results and discussion**
**3.1. Overall particle number concentrations**
Fig. 2a presents a time series of the 1 hour average total particle number concentration (CN)
measured using TSI 3776 CPC and TSI 3772 CPC throughout the sampling periods. The number
concentration of particles larger than 2.5 nm ($CN_{2.5}$) or 10 nm ($CN_{10}$) in the Arctic and Pacific marine





186 environments had a range of approximately three orders of magnitude ($\sim 10^1 - 10^3$ cm$^{-3}$). In most cases,

187 the CN$_{2.5}$ and CN$_{10}$ concentrations were less than $\sim$2000 cm$^{-3}$, with averages of 505 $\pm$ 280 and 492 $\pm$

188 264 cm$^{-3}$, respectively, which were in agreement with those reported in previous studies conducted at

189 other Arctic stations (Asmi et al., 2016;Burkart et al., 2017;Freud et al., 2017) and remote marine

190 regions (O'Dowd et al., 2014;Sellegri et al., 2006;Kim et al., 2019;Jang et al., 2019;Yum et al.,

191 1998;Hudson and Yum, 2002). For example, four years of observational data from the Arctic Climate

192 Observatory in Tiksi, Russia, showed that the monthly median CN concentration ranged from $\sim$184 cm$^{-3}$

193 in November to $\sim$724 cm$^{-3}$ in July (Asmi et al., 2016). Furthermore, Sellegri et al. (2006) reported CN

194 concentrations under clean marine sector conditions at Mace Head of a few hundreds of cm$^{-3}$ (e.g., $\sim$200

195 cm$^{-3}$ in January and $\sim$450 cm$^{-3}$ in June). Elevated CN$_{2.5}$ and CN$_{10}$ concentrations were concentrated over

196 the period from September 13 to 20, when the ship sailed over Chukchi and Bering Seas. During this

197 period, CN$_{2.5}$ and CN$_{10}$ concentrations exceeding $\sim$2000 cm$^{-3}$ were frequently observed. The peak

198 concentrations of aerosol particles were notable, as the CN$_{2.5}$ and CN$_{10}$ concentrations exceeded $\sim$6016

199 and $\sim$5750 cm$^{-3}$, respectively.

200  To elucidate further details of the variations in CN$_{2.5}$ and CN$_{10}$, the particle size distributions

201 measured with the nano SMPS, standard SMPS, and OPS were divided into four size groups: nucleation

202 (3 − 20 nm), Aitken (20 − 100 nm), accumulation (100 − 300 nm), and coarse (> 300 nm from OPS), as

203 shown in Fig. 2b−e. The average number concentrations of the nucleation-mode (N$_{NUC}$), Aitken-mode

204 (N$_{AIT}$), accumulation-mode (N$_{ACC}$), and coarse-mode (N$_{OPS}$) particles were 169 $\pm$ 142, 201 $\pm$ 131, 40 $\pm$

205 17, and 4 $\pm$ 2 cm$^{-3}$, respectively. The temporal variations in N$_{NUC}$ and N$_{AIT}$ exhibited a distinct pattern,

206 compared to that of N$_{ACC}$ and N$_{OPS}$. Overall, N$_{NUC}$ and N$_{AIT}$ concentrations larger than $\sim$1000 cm$^{-3}$ were

207 observed from September 13 to 20 (e.g. the ship sailed over Chukchi and Bering Seas), whereas

208 relatively high concentrations of N$_{ACC}$ and N$_{OPS}$ were observed from September 21 to 23 (e.g., the ship

209 sailed over Sea of Okhotsk). As shown in Fig. 2b, sudden bursts of nucleation-mode particles occurred

210 frequently, as indicated by a sudden increase in the N$_{NUC}$ concentration rising from tens to several

211 thousands of cm$^{-3}$. Whenever the CN$_{2.5}$ concentration exceeded $\sim$2000 cm$^{-3}$, the N$_{NUC}$ concentration

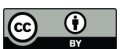



exceeded ~600 cm$^{-3}$ (except for the results observed in the evening of September 18). In addition, the
$CN_{2.5}$ concentration was strongly correlated with the $N_{NUC}$ concentration ($r^2$ = 0.69) (Fig. S3),
suggesting that the high CN concentration was mainly derived from nucleation-mode particles.
Instances of elevated $N_{NUC}$ occurred along the northern coast of Alaska (September 13 – 14, 2017),
throughout the Chukchi Sea (September 15, 2017), near the Nome and Eastern Siberia (September 16 –
18, 2017), and throughout the Bering Sea (September 19 – 20, 2017). During the cruises, the satellite-
derived chlorophyll-*a* concentration data indicated strong biological activity over the Chukchi and
Bering Seas, as shown in Fig. S1. Thus, the high occurrence of nucleation-mode particles may be
related to multiple processes that influence the formation of secondary aerosols (e.g., oceanic biological
activities, regional anthropogenic emissions on land (Alaska or eastern Siberia), and terrestrial sources
in the tundra ecosystems of Alaska).
**3.2. Case studies**

As mentioned in Section 3.1, significant increases in $N_{NUC}$ were frequently observed during the

cruise (Fig. 2 b). Typically, $N_{NUC}$ is used to indicate the presence of newly formed particles produced by
gas-to-particle conversion (i.e., secondary aerosol formation) (Asmi et al., 2016;Burkart et al., 2017).
Here, an NPF event was defined as a sharp increase in the $N_{NUC}$ with elevated $CN_{2.5}$ that lasted for at
least one hour. Fig. 3 presents contour plots of the size distributions measured using nano SMPS and
standard SMPS. This strong NPF and growth event occurred over the Chukchi and Bering Seas, which
border the western and northern sides of Alaska, suggesting that there may be a substantial source of
precursors in this region. Bursts of the smallest particles at the lowest detectable sizes (~2.5 nm) were
not observed, however, we hypothesize that, during the NPF event, particle formation occurred
elsewhere and that subsequent horizontal extension caused the particles to reach the sampling site.
Previously, NPF events have been identified on the regional scale in several locations around the world
(Kerminen et al., 2018;Németh and Salma, 2014;Vana et al., 2004;Väänänen et al., 2013). For instances,
Németh and Salma (2014) found that a nucleating air mass in regional NPF events may originate




horizontally as far as several hundreds of kilometers (~400 or 700 km) away from the sampling site. In
this section, case studies are discussed, including (i) marine Arctic NPF event, (ii) terrestrial Arctic NPF
event, and (iii) pacific marine aerosol categories. During these temporal periods, the influences of the
origins and pathways of air masses on the characteristics of particle formation and growth were
investigated.
**3.2.1. Open ocean marine Arctic NPF event case study**
The marine Arctic NPF event was observed on September 3, 2017, and time series plots of the
particle size distribution and air mass origins are presented in Fig. 4. $N_{NUC}$ increased from 77 cm$^{-3}$ to
757 cm$^{-3}$, while $N_{AIT}$ varied little. The elevated number concentration of nucleation-mode particles
lasted for over five hours and then disappeared. Geometric mean diameter (GMD) varied from 14.6 to
18.2 nm with an average of 16.3 nm, indicating that particle growth hardly occurred. During the day, air
masses traveled over the Arctic Ocean (explicitly, 47.6, 0 and 0.4 h over the Arctic Ocean, Pacific
Ocean and land domain, respectively), and have been categorized as Arctic Ocean originated air masses.
As shown in Fig. S1, the satellite-derived chlorophyll-*a* concentration indicated a relatively high level
of biological activity in the ocean during the time period focused upon in this study. It was noteworthy
that the monthly mean chlorophyll concentration in the Beaufort and Chukchi Seas (2.24 ± 3.44 mg m$^{-3}$;
65$^{o}$N−74$^{o}$N and 170$^{o}$E−120$^{o}$W) was approximately 3-fold greater than that estimated in the Pacific
Ocean including the Bering Sea and the Sea of Okhotsk (0.83 ± 1.30 mg m$^{-3}$; 40$^{o}$N−65$^{o}$N and
145$^{o}$E−168$^{o}$W) (Fig. S1). Moreover, the marginal ice zone is commonly associated with intense algae
blooms during the melting season, therefore, significant emissions of biogenic trace gases such as DMS
have been detected in the sea-ice edge (Levasseur, 2013;Oziel et al., 2017). Accordingly, as our
measurements were collected over the Arctic Ocean onboard the ice breaker, marine biogenic sources
could be considered as an important factor inducing NPF events.
Fig. 4d shows Solar Zenith Angle (SZA) data that can be used as a proxy for solar energy reaching
the ground surface. We found that the NPF event occurred when the sun was below the horizon (i.e.,



Arctic nighttime nucleation). Typically, nucleation trends to take place preferably with high solar
irradiation during the daytime (Kulmala et al., 2004). In several locations, however, also nighttime
nucleation has been observed at Tumbarumba in Australian (Suni et al., 2008), at Värriö measurement
station in Finnish Lapland (Vehkamäki et al., 2004), and at a subarctic site in northern Sweden (~14 km
east of Abisko) (Svenningsson et al., 2008). The possible explanation for nighttime events is that the
actual formation and growth occurred even during daylight, but very slow growth in the Arctic and
marine atmosphere allowed to detect the particles (~ 8 nm) only after sunset (Vehkamäki et al., 2004).
Suni et al. (2008) reported that 32% of strong nighttime nucleation events (2.5 times as frequent as
daytime nucleation event) were appeared in the presence of a very efficient ion source such as the
strong radon efflux from the Tumbarumba soil. Due to their rarity, the major mechanisms for nocturnal
aerosol production are still unclear and require more study.

**3.2.2. Open ocean terrestrial Arctic NPF event case study**
The terrestrial Arctic NPF event was observed during September 13−14 2017. As shown in Fig. 5,
significant strong NPF events occurred frequently during this period. The number concentration of total
particles increased considerably, as a $CN_{2.5}$ value exceeding ~6016 cm$^{-3}$ was observed during this event.
In addition, the average concentrations of $N_{NUC}$ and $N_{AIT}$ during the terrestrial Arctic NPF were 931 ±
222 and 1127 ± 380 cm$^{-3}$, respectively. This indicates that high $CN_{2.5}$ concentration mainly contributed
by nucleation and Aitken-mode particles (45 and 54% of the size distribution for nucleation-mode and
Aitken-mode particles, respectively). GMD increased from 13.9 to 33.3 nm, indicating that the
nucleation-mode particles subsequently increased in size. The formation and growth of aerosol particles
were observed during the daytime (Fig. 5d), suggesting that photochemistry is involved. During this
period, air masses heavily influenced by northern Alaska. The average retention times of the 2-day back
trajectories arriving at the ship position over the northern Alaska, Arctic Ocean and Pacific Ocean were
40.8, 7.2 and 0 h, respectively (Fig. 5e). It can be seen that the photochemical reactions of precursor
gases (e.g., volatile organic compounds (VOCs) such as isoprene, monoterpenes, and sesquiterpenes)





emitted by terrestrial ecosystems in Alaska were associated with new particle formation and growth
(Schollert et al., 2014;TAPE et al., 2006;Kolesar et al., 2017;Ström et al., 2003).

**3.2.3. Pacific marine aerosol case study**
A typical aerosol scenario for Pacific marine air masses was observed on September 21−22, 2017,
when the air masses passed over mainly the Pacific Ocean (including the Bering Sea and Sea of
Okhotsk) (explicitly, 0, 47.9 and 0.1 h over the Arctic Ocean, Pacific Ocean and land domain,
respectively) (Fig. 1a). As shown in Fig. 6, the aerosol number concentrations exhibited a bimodal size
distribution, peaking at size ranges of $30 − 80$ nm (Aitken mode) and $100 − 300$ nm (accumulation
mode), respectively. In contrast, the concentrations of nucleation-mode particles were very low. For
example, the concentration of $N_{NUC}$ ranged from 1 to 38 $cm^{-3}$ with an average of $8 \pm 4$ $cm^{-3}$. We also
observed $CN_{2.5}$ values at the background level of $\sim460 \pm 70$ $cm^{-3}$, which are consistent with the
measurements collected at a coastal Antarctic station during summer ($\sim600$ $cm^{-3}$) (Kim et al., 2017) and
from flight-based measurements over the Arctic Ocean ($\sim300$ $cm^{-3}$) (Burkart et al., 2017).

**3.3. Overview of aerosol properties according to different air mass back trajectories**
Air masses comprising marine Pacific along with marine and terrestrial Arctic air messes were
encountered during the campaign. In the section 3.2, two case studies of NPF events (Fig. 4 and Fig. 5)
were found in the Arctic atmosphere. As stressed in Willis et al., (2018), NPF and growth is frequently
observed in the boundary layer in the both Arctic open ocean and coastal regions. These events seem to
occur more frequently than lower-latitude marine boundary layers (Quinn and Bates, 2011); there are
multiple reasons including summer 24-h high solar radiation, low condensation sink, low temperature
and low mixing of surface emissions, as recently reviewed in Abbatt et al. (2019). Our study also
confirmed that any NPF was not detected during the Pacific transect.
In this section, we present an overall meteorological air mass summary of the open ocean field study,
categorizing it into three synoptic period types: Pacific marine, Arctic marine and Arctic terrestrial.



These classifications do not represent specific air mass back trajectories analysis, but they can mainly
represent air masses that have been travelled over these three distinct geographical regions (section 2.4).
Average size distributions for the three selected periods in the different air masses are shown in Fig. 7.
In addition, a summary of total number concentrations of particles for these periods is included in Table

1.


-   *Arctic Marine*. A trimodal distribution was seen at 18 ± 3 nm, 53 ± 6 nm and 150 ± 6 nm. The first
mode is due to NPF arriving from open pack sea ice and open ocean Arctic regions, as discussed in
Section 3.2.1 where a case study is presented. The Aitken mode (~53 nm) is remarkably similar to the
Pacific Ocean aerosol size distribution and to previous studies detected in the Arctic regions (Tunved et
al., 2013;Freud et al., 2017;Dall'Osto et al., 2019). The largest mode at ~150 nm may be due to a
combination of primary and secondary aerosol components.

- *Arctic terrestrial*. A bimodal distribution is seen with two main modes at 24 ± 3 nm and 151 ± 3 nm,
respectively. The nucleation and Aitken modes are much higher than the accumulation mode, suggesting
that NPF governs the aerosol processes in this coastal region at this time of the year.

-   *Pacific marine*. The Pacific Ocean aerosol size distributions showed a trimodal size distribution at 56
± 3 nm, 130 ± 3 nm and 220 ± 6 nm. The lowest peak at ~56 nm (i.e., Aitken mode) is likely a
combination of primary and secondary marine aerosol components, whereas the largest peak at ~220
nm might be caused by cloud processing and aged aerosols. The mode at ~130 nm could originate from
primary sea spray aerosols in the Pacific atmosphere (Quinn et al., 2015). When the distribution is fitted
with log-normal modes, the inter-modal minimum is calculated to be ~120 nm - often known as Hoppel
minimum as a signature of cloud processing (Hoppel et al., 1994) - although, it is difficult to draw a
firm conclusion due to the overlap with the third mode at ~130 nm.

This study shows that aerosol originating from higher and lower marine latitudes – although both



being treated as marine air masses - have very different features, as pointed out in several previous

studies (Dall'Osto et al., 2010;Frossard et al., 2014). A key conclusion of this study is that we also need

to separate different bioregions in the Arctic, especially given the current results showing very different

aerosol size distributions in the Arctic study areas (Fig. 7; Arctic marine and Arctic terrestrial). The

reasons for the much higher aerosol concentrations near the coast of Alaska relative to the open ocean

sympagic and pelagic regions may be multiple. We discuss at least two major sources may contribute to

the high aerosol concentrations recorded.

The first source of aerosols in the terrestrial Arctic air masses may be due to anthropogenic sources.

Due to sea ice retreat and better technologies, the Arctic is now easily accessible to human activities,

including oil and gas extraction (Law and Stohl, 2007;Peters et al., 2011). These Arctic oil fields can

emit the large amounts of aerosols, and with on-going Arctic development, such local combustion

emissions may increase in the future, possibly affecting local air quality (Gunsch et al., 2017;Schmale et

al., 2018a). In fact, some NPF events were reported within the North Slope of Alaska (e.g., Prudhoe Bay

oil fields) during August and September 2016 at Oliktok Point Alaska. This observation was suggested

to be linked with oil fields emissions (Kolesar et al., 2017). However, our measurements were

conducted in the open ocean, quite far from any land oil field local emissions. BC data were collected as

shown in Fig. 8; they revealed very high standard deviations due to high detection limit of the

instrument used relative to the concentrations detected. However, no remarkable differences can be seen,

all pointing to pristine clean marine air masses with BC values of approximately $20 \pm 10$ ng m$^{-3}$. The

two Arctic categories (Marine and Terrestrial) shows similar BC values, whereas higher values can be

seen for the Pacific marine aerosol category, probably due to contamination from nearby Asian high

pollutant sources.

The second source of aerosol in the terrestrial Arctic air masses may be due to terrestrial natural

sources. We believe that this may be a much more probable reason. The Arctic Ocean is submerged

under areas of relatively shallow water known as a shelf sea for ~50% of its area. It is a relatively small

ocean, characterized by pronounced riverine influence and a complex hydrography. Up to 11% of the





entire global river discharge ends up in the Arctic Ocean (Shiklomanov et al., 2000), which is only 1%
of the global ocean volume. The discharge of freshwater is increasing (Peterson et al., 2002), impacting
coastal salinity and carbon cycle. Indeed, this continental runoff is a major source of freshwater,
nutrients and terrigenous material to the Arctic Ocean (Benner et al., 2005;Fichot et al., 2013;Massicotte
et al., 2017). The warming climate in the region is causing permafrost degradation, alterations to
regional hydrology and shifting amounts and composition of dissolved organic matter (DOM)
transported by streams and rivers (Mann et al., 2016;Chen et al., 2017). Overall, there is a considerable
spatial and temporal heterogeneity in the distribution of the DOC in the Arctic, owing to strong
biological and physicochemical processes. It is important to remember that sea ice formation and
melting also affects the concentrations and distributions of DOC, although its impact is still difficult to
resolve (Fichot et al., 2013;Shen et al., 2012).
In a recent paper (Park et al., 2019), we suggested that the large amount of freshwater from river
runoff may have a substantial impact on primary aerosol production mechanisms, possibly affecting the
cloud radiative forcing. We showed that the Artic riverine organic matter can be directly emitted from
surface seawater into the atmosphere via bubble bursting (Park et al., 2019). The high amount of DOC
populating the sea-surface microlayer (SML) in the Arctic waters - including UV absorbing humic
substances - can also produce VOCs (Ciuraru et al., 2015;Fu et al., 2015), which are known precursors
of secondary organic aerosols. Recently, Mungall et al. (2017) reported that the marine microlayer in the
Canadian Arctic Archipelago is a source of oxidized VOCs (OVOCs), which could be an important
source of biogenic secondary organic aerosol (Croft et al., 2019). Previous studies also reported
fluorescent water-soluble organic aerosols in the High Arctic atmosphere (Fu et al., 2015). It is worth
noting that terrestrial VOCs from tundra and lakes at elevated concentrations were reported (Potosnak et
al., 2013;Lindwall et al., 2016;Steinke et al., 2018).
Fig. 9 shows DOC concentrations from water samples taken in the areas where the NPF marine and
terrestrial case studies (Section 3.2.1 and 3.2.2) were detected. It is clear that as much as twice higher
concentrations are seen for the coastal marine areas, relative to the open ocean marine regions. The


origin of this organic matter can be obtained by the FDOM analysis. Fig. 9 (bottom) shows specific
peaks attributed to different chemical features. The ratio of terrestrial humic substances (peak A) was
3.5 for the terrestrial/marine samples. By striking contrast, marine fulvic substances (peak M) and
proteinaceous (peak T) had a ratio of 0.45 and 0.27, respectively, showing two very distinct chemical
compounds. This suggests that coastal oceanic water enriched in river organic material as well as fresh
water tundra and lake may be a source of VOC (both from biotic and abiotic emission processes) that
may be responsible for the high secondary aerosols detected near these areas.
**3.4. Impact on CCN number concentrations**
Fig. 10a illustrates the CCN concentrations for the three selected periods under different
supersaturation conditions. For a given SS of 0.4%, CCN concentrations for Arctic marine, Arctic
terrestrial and Pacific marine air masses were $35 \pm 40$ cm$^{-3}$, $71 \pm 47$ cm$^{-3}$, and $204 \pm 87$ cm$^{-3}$,
respectively. Higher concentrations of CCN were observed when the air mass originated from the
Pacific marine for a SS of 0.2%−1.0 %. This may have occurred due to the differences in the CCN
sources between the Arctic and Pacific Oceans. It was noted that that accumulation and coarse-mode
particles, which are predominant over the Pacific Ocean (Fig. 7), can easily act as CCN. Our results
agreed well with values reported in previous studies that measured CCN at a ground-based Arctic
station (Jung et al., 2018), but was slightly higher than those measured from high-Arctic expeditions
(Leck et al., 2002;Martin et al., 2011;Mauritsen et al., 2011). For example, Jung et al. (2018) reported
seasonal variations in the CCN concentration over seven years (2007 −2013) at the Zeppelin station, and
found that the monthly mean CCN concentrations ranged from 17 cm$^{-3}$ in October 2007 to 198 cm$^{-3}$ in
March 2008 at a SS value of 0.4%. However, Mauritsen et al. (2011) observed CCN concentrations
lower than ~100 cm$^{-3}$ at five different supersaturations (SS = 0.10%, 0.15%, 0.20%, 0.41%, and 0.73%),
with median values ranging from 15 to 50 cm$^{-3}$, in four High Arctic expeditions during the Arctic
Summer Cloud Ocean Study. Such values were also in line with the long term measurement at an Arctic
station in Barrow, which indicated that the median CCN concentrations at 0.2% SS was smaller than





100 cm$^{-3}$ (Schmale et al., 2018b).
We also compared CCN activity and critical diameter for the three selected periods, as shown in Fig.
10b and c. The CCN activity is defined as the ratio of the number concentration of particles that
activated to become CCN at a given supersaturation to the total number concentration of particles larger
than 2.5 nm (CN$_{2.5}$). The CCN activity followed a similar pattern as the CCN concentration.
Furthermore, the critical diameter ($D_c$) was estimated using the measured aerosol size distribution,
CN$_{2.5}$, and CCN concentrations with a time resolution of 1 h, as described by Furutani et al., (2014).
The $D_c$ at a SS of 0.4% was found to be 103 ± 43 nm, 83 ± 18 nm, and 136 ± 67 nm for Arctic marine,
Arctic terrestrial, and Pacific marine periods, respectively. These values are comparable to previous
studies obtained in the Arctic and subarctic regions. For instance, Jaatinen et al. (2014) reported that the
$D_c$ value of 98 ± 16 nm (SS = 0.4%) from the subarctic area of Finland (Pallas-Sodankylä Global
Atmospheric Watch station). Anttile et al. (2012) also showed that a $D_c$ value was in the range of 90 to
120 nm at a SS of 0.4% during the same filed campaign as reported in Jaatinen et al. (2014). For a
maximum SS between 0.18 and 0.26%, $D_c$ varied between 110 and 140 nm at the same measurement
sites.
**4. Summary and conclusions**
This study presents the physical properties of aerosol particles measured aboard the R/V Araon ice-
breaker during 2017 throughout the Arctic and Pacific Oceans. The CN$_{2.5}$ value commonly ranged
between 13 and 2,000 cm$^{-3}$ with an average of 505 ± 280 cm$^{-3}$. An elevated CN$_{2.5}$ concentration
reaching ~6,016 cm$^{-3}$ was observed from 13 September to 20 September. The temporal variations in the
CN$_{2.5}$ concentration followed a similar pattern to those of N$_{NUA}$ and N$_{AIT}$. We also found that the CN$_{2.5}$
concentration was strongly correlated with N$_{NUA}$ ($r^2$ = 0.69), suggesting that CN was mainly derived
from nucleation-mode particles.
NPF events caused by gas-to-particle conversion frequently occurred over the Arctic Ocean.
Overall, two major NPF sources (i.e., Arctic marine and Arctic terrestrial) were identified based on the



backward air mass trajectory analysis. NPF events were associated with Arctic marine air masses,
indicating the impact of marine biogenic emissions from the Arctic Ocean. Strong NPF events with
particle growth were associated with Arctic terrestrial air masses, which may be due to the biogenic
precursor gases emitted by terrestrial ecosystems including river discharge and Alaskan tundra in the
Arctic coastal areas. In contrast, relatively larger particles with broad Aitken and accumulation-mode
peaks were observed over the Pacific Ocean. Our study confirmed that any NPF was not detected during
the Pacific transect. We also compared the average CCN concentrations for each of the cases. Our data
showed that the impact of aerosols on CCN concentrations (SS = 0.4%) was significant: $35 \pm 40$ cm$^{-3}$,
$71 \pm 47$ cm$^{-3}$, and $204 \pm 87$ cm$^{-3}$ for Arctic marine, Arctic terrestrial, and Pacific marine periods,
respectively. Our interpreted data showed that river outflows and tundra strongly influence Arctic
aerosol properties. Further detailed measurements of the chemical characteristics of marine aerosols are
required to provide more direct evidence for the contribution of biogenic precursors to the NPF and
CCN in the remote Arctic atmosphere.
Arctic areas are currently experiencing drastic climate change, with air temperatures increasing at
twice the rate of the global average. This warming is causing clear changes, such as the increases in
biogenic emissions from tundra vegetation and changes in vegetation cover (Faubert et al.,
2010;Peñuelas and Staudt, 2010;Potosnak et al., 2013;Lindwall et al., 2016). Lindwall et al. (2016)
observed a 280% increase in VOC emissions relative to the ambient level in response to a 4 ℃ increase
in the summer temperature of the Subarctic. Increases in VOC emissions from river discharge and
tundra vegetation in the Arctic are critical factors that induce NPF and particle growth events, which
may impact the CCN concentrations during the Arctic summer.

**Data availability**
The data analyzed in this publication will be readily provided upon request to the corresponding author
(yjyoon@kopri.re.kr).

**Author contributions**



JP, YJY designed the study, JP, MD'O, KP, YG, HJK, EJ, KTP, MP, SSY, JJ, and BYL analyzed data.

JP, MD'O, KTP and YJY prepared the manuscript with contributions from all co-authors.

**Competing interests**

The authors declare that they have no conflict of interest.

**Acknowledgements**

We are grateful to the captain and crews of R/V *Araon* for their enthusiastic assistance during the cruise of ARA08C. This work was supported by a Korea Grant from the Korean Government (MSIP) (NRF-2016M1A5A1901769) (KOPRI-PN19081) and the KOPRI projects (PE17390). Kihong Park was supported by the National Leading Research Laboratory program (NRF-2019R1A2C3007202). Minsu Park and Seong Soo Yum were supported by National Research Foundation of Korea (NRF) grant (NRF-20180R1A2B2006965). Jinyoung Jung was supported by "Korea-Arctic Ocean Observing System (K-AOOS)", KOPRI, 20160245, funded by the MOF, Korea.

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






Table 1. A summary of total number concentrations of particles measured with TSI 3776 CPC, TSI
3772 CPC, Standard SMPS, and nano SMPS for the three selected periods.

|  | Pacific Ocean | Arctic Marine | Arctic Terrestrial |
|---|---|---|---|
| Periods | 9/21/2017−9/23/2017 | 9/02/2017−9/05/2017, 9/10/2017−9/12/2017 | 9/13/2017−9/17/2017 |
| $CN_{2.5}$ | $397 \pm 185$ cm$^{-3}$ | $413 \pm 442$ cm$^{-3}$ | $1622 \pm 1450$ cm$^{-3}$ |
| $CN_{10}$ | $384 \pm 86$ cm$^{-3}$ | $414 \pm 452$ cm$^{-3}$ | $1396 \pm 1279$ cm$^{-3}$ |
| $CN_{2.5\text{-}10}$ | $35 \pm 195$ cm$^{-3}$ | $62 \pm 130$ cm$^{-3}$ | $263 \pm 318$ cm$^{-3}$ |
| $N_{\text{Standard SMPS}}$ | $224 \pm 83$ cm$^{-3}$ | $204 \pm 215$ cm$^{-3}$ | $739 \pm 819$ cm$^{-3}$ |
| $N_{\text{nano SMPS}}$ | $117 \pm 234$ cm$^{-3}$ | $159 \pm 194$ cm$^{-3}$ | $749 \pm 864$ cm$^{-3}$ |






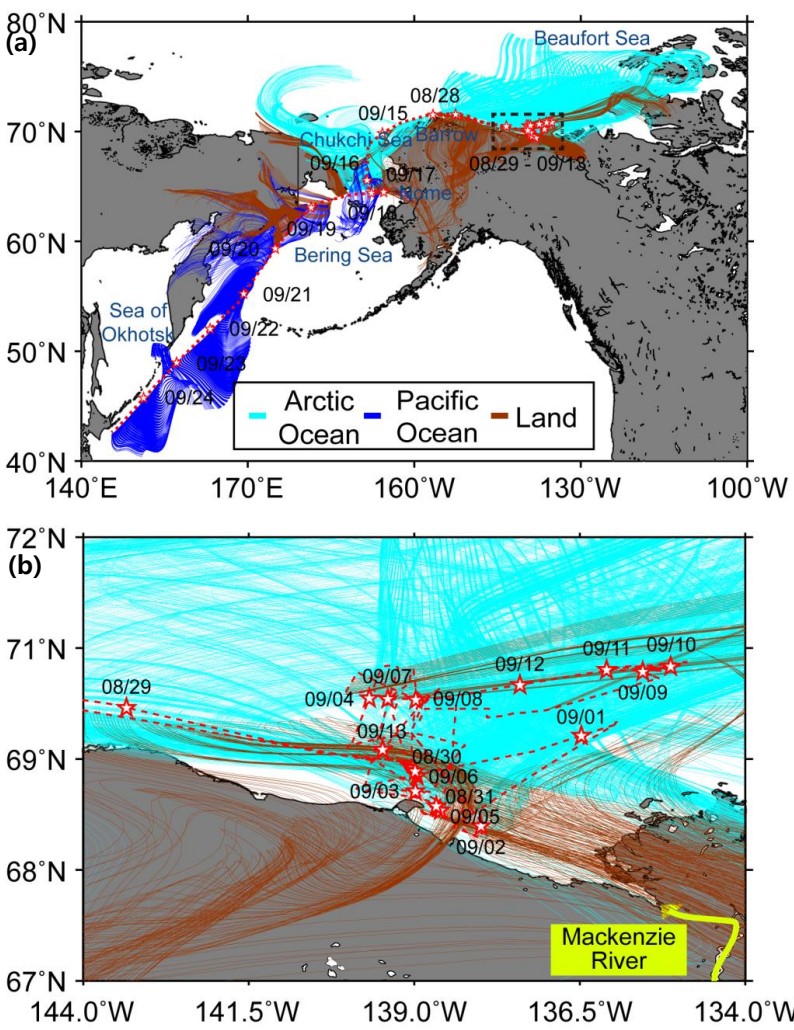


Figure 1. Ship tracks across (a) the Arctic (8/28/2017−9/18/2017) and Pacific Oceans

(9/18/2017−9/25/2017) and (b) zoom into the dotted black square region in Fig. 1a. A dotted red line

including star symbols represents ship tracks during the entire cruise. The star symbols represent the

daily ship location at midnight. Light blue, blue and brown lines denote the 2-day air mass trajectories

categorized into three main domains such as Arctic Ocean, Pacific Ocean, and land, respectively.

815

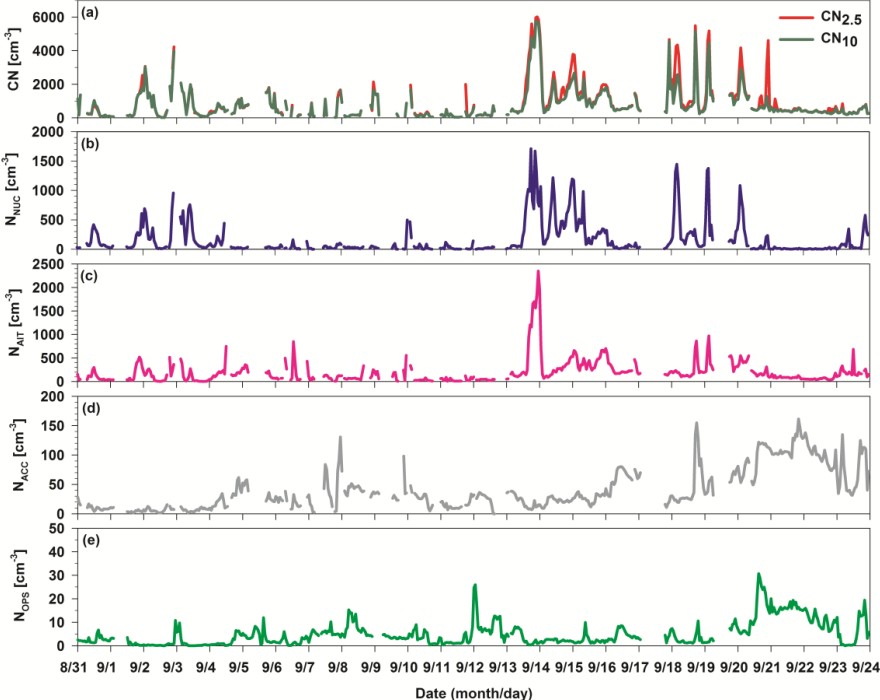

816

Figure 2. Time series of the 1 hour average (a) total aerosol ($CN_{2.5}$ and $CN_{10}$), (b) nucleation-mode (3 −

20 nm) ($N_{NUC}$), (c) Aitken-mode (20 − 100 nm) ($N_{AIT}$), (d) accumulation-mode (100 − 300 nm) ($N_{ACC}$),

and (e) coarse-mode (> 300 nm from OPS) ($N_{OPS}$) number concentrations. The $CN_{2.5}$ and $CN_{10}$

represent total number concentration of particles larger than 2.5 and 10 nm, respectively.




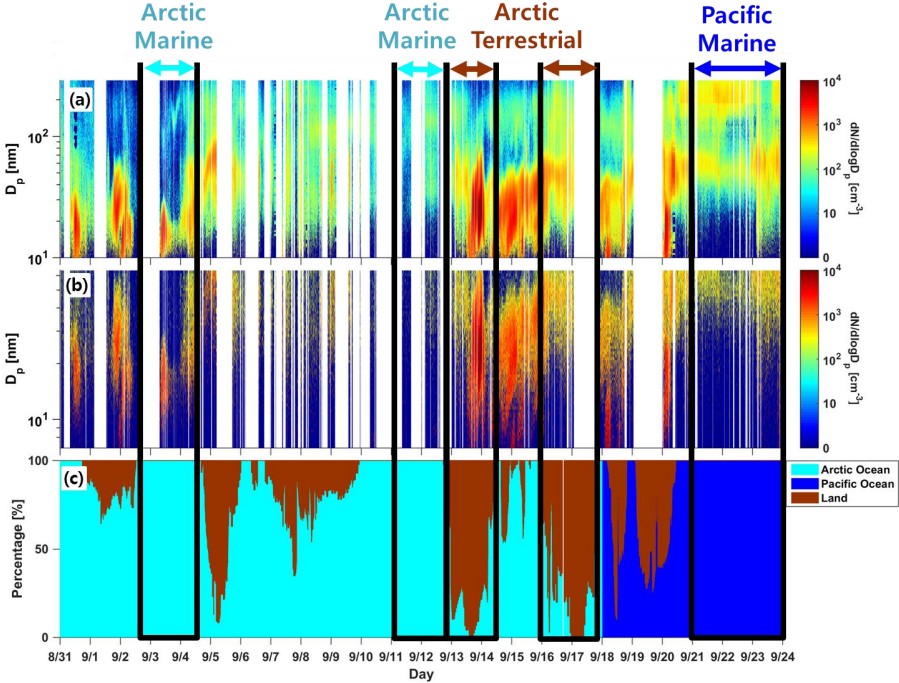


Figure 3. Contour plots of the size distributions measured using (a) standard and (b) nano SMPS and (c)
the residence time of air masses that passed over the Arctic Ocean, Pacific Ocean, and land throughout
the sampling periods.




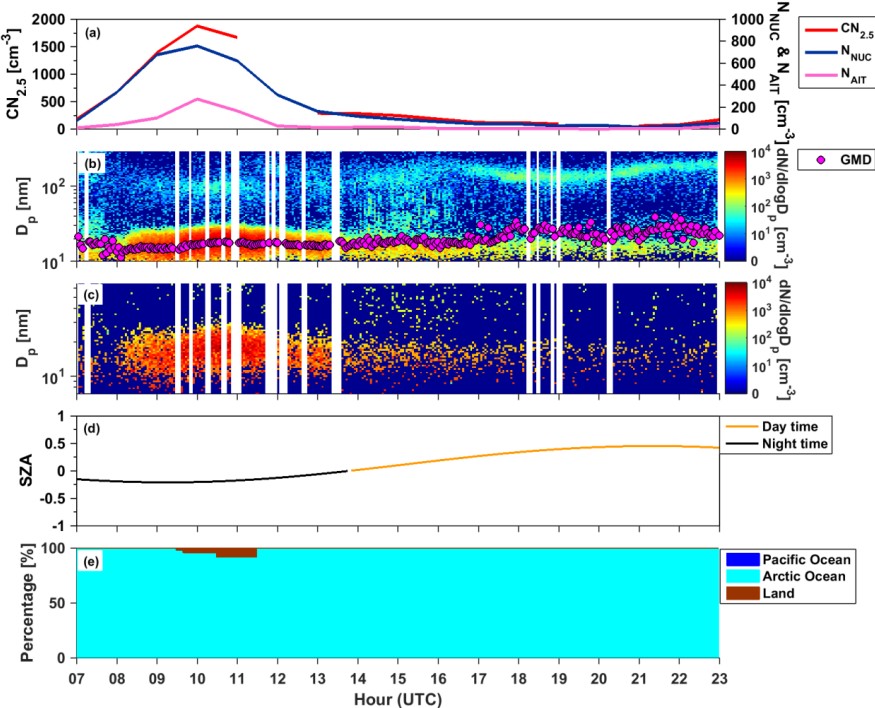


Figure 4. Example of a case-I event observed on 3 September 2017. From top to bottom, the parameters

are: (a) the total number concentration of particles smaller than 2.5 nm, nucleation-mode particles, and

Aitken-mode particles; (b) a time series of the standard SMPS size distribution and GMD; (c) a time

series of the nano SMPS size; (d) Solar Zenith Angle; (e) the residence time of air masses that passed

over the ocean, land, and sea-ice areas.

835

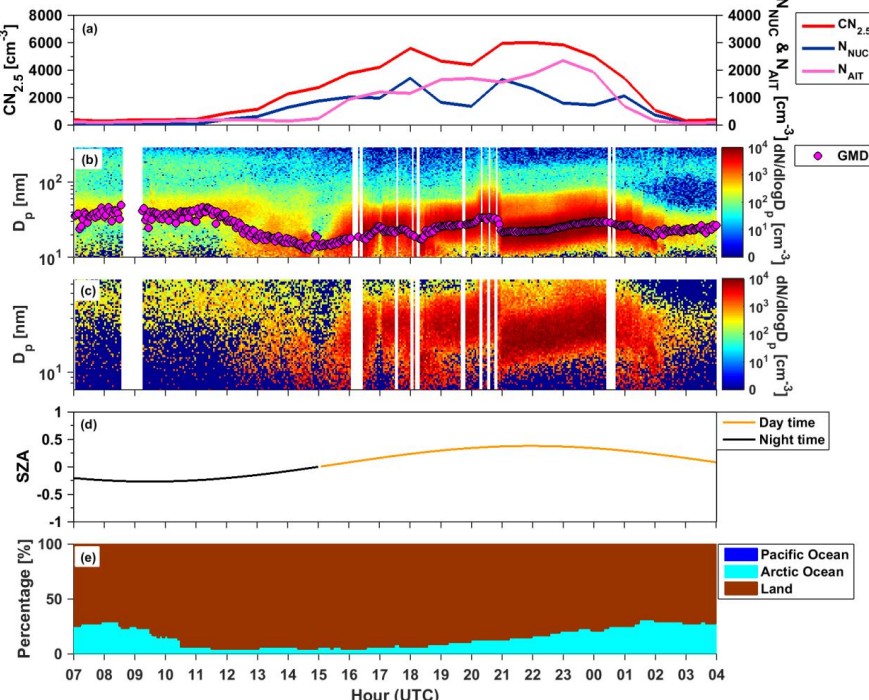

836

Figure 5. Example of a case II event that was observed on September 13−14, 2017. From top to bottom,

the parameters are: (a) the total number concentration of particles smaller than 2.5 nm, nucleation-mode

particles, and Aitken-mode particles; (b) a time series of the standard SMPS size distribution and GMD;

(c) a time series of the nano SMPS size; (d) Solar Zenith Angle; (e) the residence time of air masses that

passed over the ocean, land, and sea-ice areas.


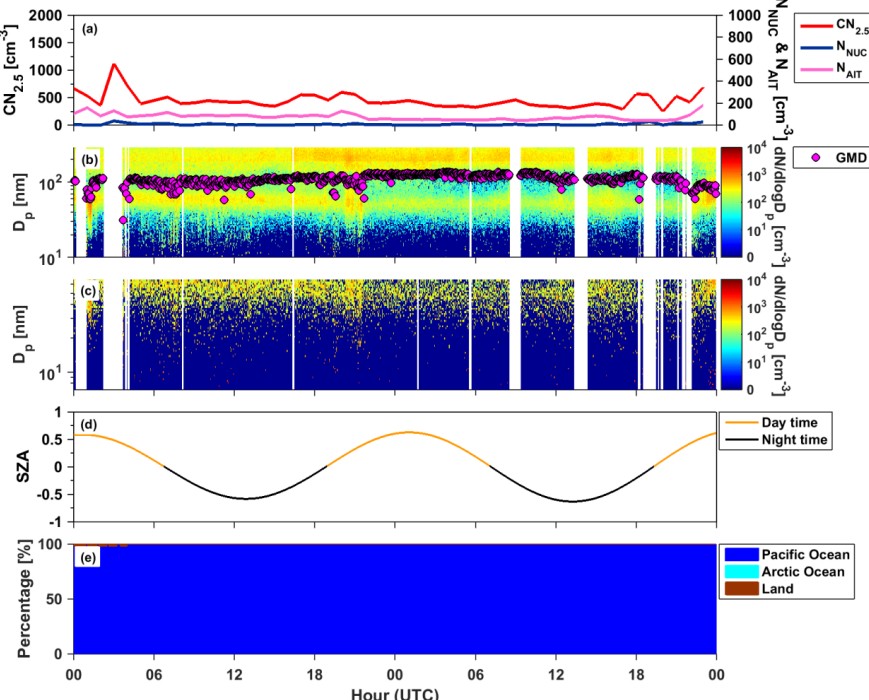


Figure 6. Example of a case III event that was observed on September 21−22 2017. From top to bottom,
the parameters are: (a) the total number concentration of particles smaller than 2.5 nm, nucleation-mode
particles, and Aitken-mode particles; (b) a time series of the standard SMPS size distribution and GMD;
(c) a time series of the nano SMPS size; (d) Solar Zenith Angle; (e) the residence time of air masses that
passed over the ocean, land, and sea-ice areas.





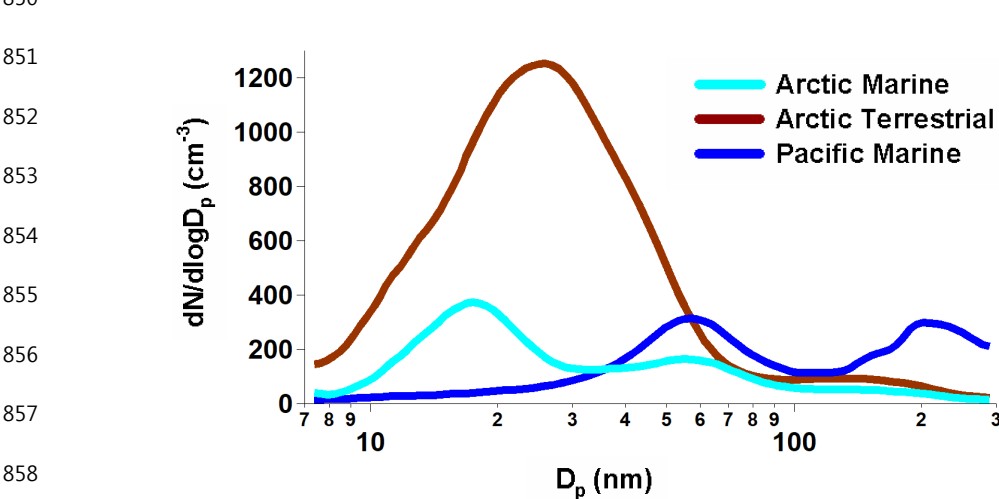

Figure 7. Average size distributions of aerosol particles for Arctic marine, Arctic terrestrial and Pacific

marine air masses




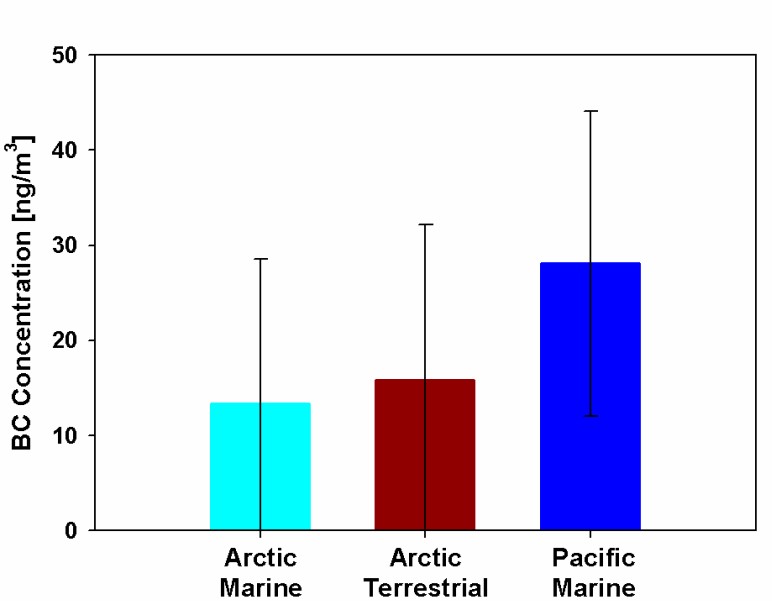


Figure 8. Average mass concentrations of black carbon for each air mass.






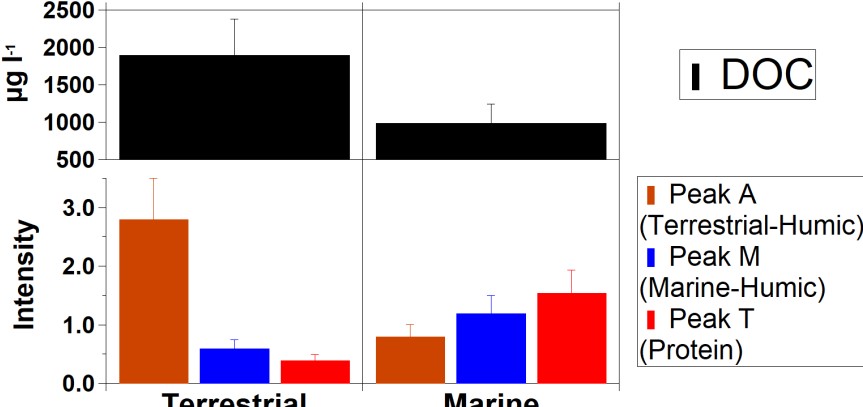


Figure 9. Average DOC concentrations for surface seawater samples collected during this cruise,

simultaneously during the atmospheric measurements herein reported. Peak A, M, and T represent

terrestrial-humic substances, marine-fulvic substances, and protein, respectively.







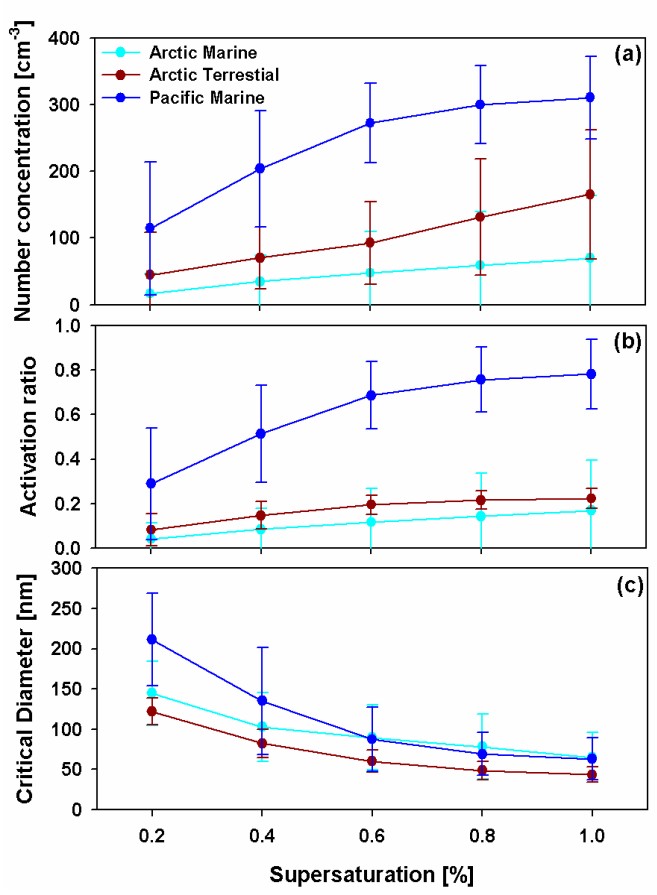


Figure 10. Comparisons of (a) CCN number concentrations, (b) CCN activity, and (c) critical diameter
for Arctic marine, Arctic terrestrial and Pacific marine air masses under different supersaturation
conditions. The error bars represent a standard deviation.