# Peer review of "Shipborne observations reveal contrasting Arctic marine, Arctic terrestrial and Pacific marine aerosol properties"

_Atmospheric Chemistry and Physics, 2019_

## Referee Comment (RC1) · Anonymous Referee #1 · 9 Jan 2020

The manuscript deals with marine aerosol physical properties retrieved during a late summer vessel cruise in Arctic and Pacific Ocean, and it's really interesting, finding out different and peculiar aerosol properties, allegedly due to origin/sources, with a clear distinction between well defined region (Arctic Marine, Arctic Terrestrial and Ocean Pacific. The main result is the relevance of terrestrial ecosystem in affecting the aerosol physical properties, at least for late summer period.

going directly to the paper:

the title and the abstract are clear and reflect the paper's content, the text is precise, fluent and well written there is only a simple error when authors describe relationship

between NPF event and photochemistry: figures 4d, 5d and 6d refers to graph with SZA vs time, but the figure (and the text) indeed show cos(SZA): please modify text and figure caption.

by the way, the figure show the "geometric solar configuration", indicating cos(SZA) as a proxy for the solar energy reaching the surface. Clouds usually affect the real solar radiation reaching the surface, while the figures show a modeled clear sky condition. could authors add any comment about eventual cloud presence and their influence on NPF event?

some comment about the text:

line 84: figure 2 shows data analysis between August 31 and September 24: please change data in line 84

line 122-123: for each instrument is indicated the data frequency sampling, except the OPS(TSI3330): could the authors supply this information?

line 233: maybe a ";" should replace "," between observed and however

line 248: could authors add a definition of Geometric Mean Diameter?

line 285-286: maybe a verb is missing?

line 350: i would add the word "late summer" before terrestrial line 365 : the same here

line 426: could authors add a definition of Critical Diameter?

simple typos: line 306: masses instead of messes

line 440 and 441 , in these lines the authors use a "," to indicate number greater tha 1000, while in manuscript generally no comma is used. please adjust the numbers line 270-271 : please adjust references: this format (Vehkamäki et al., 2004) or this format? Suni et al. (2008)

several lines: when indicating multiple references, please add a space after ","

about figure: I suggest to add the aerosol origin classification , as showed in figure 3, also in figure 2.

in figure 7 the averaged size distribution are showed, starting from more than 7 nm to 300 nm. Could the author describe how the 2 smps dataset (nano and standard) are combined? The nano measures from 3 nm to 80 nm, while the standard collect data from 10 nm to 300 nm.

---

## Referee Comment (RC2) · Anonymous Referee #2 · 15 Jan 2020

This manuscript describes approximately one month of measurements of aerosol physical properties and CCN concentrations taken aboard ship in the Pacific region of the Arctic, and Northern Pacific Ocean. This work demonstrates that terrestrial inputs in coastal areas can be an important driver of the particle size distribution in Arctic regions, and can drive strong new particle formation relative to open ocean areas of the Pacific. This is an important and interesting conclusion that has not been made in the past. Also, notably, this work presents measurements of particle size distributions starting at 3nm, which has not been done in many Arctic regions. This manuscript is well written, and presents the main results clearly. I support publication in ACP once the following issues can be addressed.

[Figure]

**Major comments:**

(1) I am somewhat surprised that the authors have not demonstrated agreement between the two SMPS systems in their overlapping size range (i.e., 10-80nm). This is a crucially important size range for understanding particle growth and adequate quality control of these measurements should be addressed. How does N10-80 measured by the two instrument compare across the measurement period? Were any size calibrations of the two instruments made in the field? A clearer description of how the two measurements of particle size were aggregated over the overlapping range to produce Figure 7 is needed.

(2) How does the condensation sink and average growth rate compare (1) across the different types of air masses sampled, and (2) to previous Arctic measurements? Was the condensation sink significantly higher in the Pacific air masses? Figure 7 would suggest that is the case. I suggest that the authors make full use of their unique data set by calculating the above two quantities wherever possible, and including them in relevant figures. Statistics of these quantities could be included in Table 1. This would facilitate improved comparison with other studies in similar coastal Arctic regions (e.g., Collins et al., ACP, 2017 https://doi.org/10.5194/acp-17-13119-2017 and Burkart et al., GRL, 2017 doi: 10.1002/2017gl075671). In particular, Collins et al 2017 provides a detailed comparison of the condensation sink in their observations with that sampled globally (their Figure 8). Placing the author's results in this broader context would strengthen the paper significantly.

(3) What other environmental variables were different between the three air mass source regions? e.g., wind speeds, cloud cover?

(4) It appears as though multiple modes were present in some cases (e.g., during 9/13-9/17), and based on Figure 7 appear in average size distributions. As discussed in Burkart et al., GRL, 2017 (doi: 10.1002/2017gl075671), comparing the growth rates of different particles modes can provide significant insight into the physical chemical

properties of the condensing/partitioning species. I do not suggest that the authors undertake a detailed modelling exercise as in Burkart et al, rather that the authors calculate growth rates for the different modes and discuss what these quantities might mean in the context of the Burkart et al analysis.

**Minor comments:**

L67-68: I agree with this statement in general; however, the authors miss two very relevant studies of ship-based coastal Arctic measurements: Collins et al., ACP, 2017 https://doi.org/10.5194/acp-17-13119-2017 and Burkart et al., GRL, 2017 doi: 10.1002/2017gl075671

L69-82: This paragraph focuses on studies that have linked NPF events to increases in measured CCN; however, the authors focus not only on studies from Arctic regions and seem to focus on global observations as the expense of being exhaustive in discussing all of the very few relevant Arctic studies. The two studies mentioned in the comment above should be included here, and are more relevant to the authors' discussion than Kalivitis 2015 and Rose 2017. Burkart et al., ACP, 2017 (doi: 10.5194/acp-17-5515-2017) also connect NPF with CCN in coastal Arctic environments.

L123-126: Time resolution of the aethelometer?

L141: Can the authors be more quantitative in describing "particularly high (spike) and varied dramatically in a short time"?

L235: Tremblay et al, ACP, 2019 (https://doi.org/10.5194/acp-19-5589-2019) demonstrated regional growth events taking place over the Northern Canadian Arctic Archipelago, and their results are relevant to this discussion.

L262-263: Could a satellite-based measurement of cloud fraction (e.g., from Aqua MODIS that is openly available) be used to assess any impact of cloudiness on the observations of NPF?

L422-435: Burkart et al., ACP, 2017 (doi: 10.5194/acp-17-5515-2017) also estimated

the CCN activation diameter to be approximately 80nm in a coastal Arctic environment.

Table 1: Do NStandardSMPS and NnanoSMPS include the overlapping size range? I suggest that the authors be explicit about the size range here, rather than referring to the instrument. Presenting mean concentrations in specific size ranges from aggregated size distribution data may be more useful (e.g., as shown in figure 2)

Figures 3-6: Could more than one tick label be added to the size scale on the nanoSMPS size distributions?

Figure 7: Specify which SMPS is used in the size range from 10-80nm

**Specific comments:**

L61: Ice and snow-covered regions

L264: change 'trends' to 'tends'

L286: missing 'were'

---

## Author Response (AR1)

The manuscript deals with marine aerosol physical properties retrieved during a late summer vessel cruise in Arctic and Pacific Ocean, and it's really interesting, finding out different and peculiar aerosol properties, allegedly due to origin/sources, with a clear distinction between well defined region (Arctic Marine, Arctic Terrestrial and Ocean Pacific. The main result is the relevance of terrestrial ecosystem in affecting the aerosol physical properties, at least for late summer period.
We would like to thank the reviewer for valuable and constructive comments and suggestions. Below, you will find a point by point description of how each comment was addressed in the manuscript.

going directly to the paper:
the title and the abstract are clear and reflect the paper's content, the text is precise, fluent and well written there is only a simple error when authors describe relationship between NPF event and photochemistry: figures 4d, 5d and 6d refers to graph with SZA vs time, but the figure (and the text) indeed show cos (SZA): please modify text and figure caption.
Response: We agree with the reviewer's comment. It was modified in text and figure.

by the way, the figure show the "geometric solar configuration", indicating cos(SZA) as a proxy for the solar energy reaching the surface. Clouds usually affect the real solar radiation reaching the surface, while the figures show a modeled clear sky condition. could authors add any comment about eventual cloud presence and their influence on NPF event?
Response: To classify this issue, we compared weekly averaged liquid cloud fraction from Aqua/MODIS retrieval data for each period (week 1: 8/29/2017− 9/5/2017, week 2: 9/6/2017− 9/13/2017, week 3: 9/14/2017− 9/21/2017, and week 4: week 4 (9/22/2017− 9/29/2017)). Then, Figure S5 was added and the sentence (including references) was rewritten as given below.

Page 11, Line 270: "In addition, cloudiness which usually affects the real solar radiation reaching the surface was compared based on Moderate Resolution Imaging Spectroradiometer (MODIS) cloud fraction retrievals (Fig. S5). The data showed that cloud fraction was significantly high during the entire sampling periods, in general agreement with some other studies over the western Arctic region (e.g., Dong et al., 2010; Collines et al., 2017). In detail, the cloud fraction was relatively low for week 1 (8/29/2017− 9/5/2017; Fig. S5a) and week 3 (9/14/2017− 9/21/2017; Fig. S5c) when NPF event and growth was frequently observed (Fig. 3). This suggests that solar radiation at the surface, which is affected both by the cloud cover and SZA, may have influenced aerosol concentration and NPF observed here."

some comment about the text:
line 84: figure 2 shows data analysis between August 31 and September 24: please change data in line 84

Response: Thank you for the comment. It was corrected.

line 122-123: for each instrument is indicated the data frequency sampling, except the OPS(TSI3330): could the authors supply this information?
Response: We added the information as follows:

Page 5, Line 122: "An OPS (TSI 3330) was used to determine the size distribution of particles in the range of 100 nm to 10 μm diameter with a sample flow rate of 1.0 lpm every 3 minutes."

line 233: maybe a ";" should replace "," between observed and however
Response: It was altered.

line 248: could authors add a definition of Geometric Mean Diameter?
Response: We added the information (including the reference) as given below.

Page 10, Line 254: "The GMD is defined as the particle diameter at which the cumulative probability becomes 50% for the fitted log-normal distribution (Hinds, 1999)".

line 285-286: maybe a verb is missing?
Response: Thank you for pointing this out. It was added as follows.

Page 12, Line 300: "During this period, air masses were heavily influenced by northern Alaska."

line 350: i would add the word "late summer" before terrestrial line 365: the same here
Response: Following the reviewer's suggestion, "terrestrial Arctic air masses" were changed to "late summer terrestrial Arctic air masses" (Line 368 and Line 383).

line 426: could authors add a definition of Critical Diameter?
Response: We included the information as given below.

Page 18, Line 472: "Furthermore, the critical diameter ($D_c$) (i.e., the diameter at which the integration of aerosol size distribution from the largest particle diameter to the lower ones matches with the measured CCN concentration) was estimated using the measured aerosol size distribution, CN2.5, and CCN concentrations with a time resolution of 1 h, as described by Furutani et al., (2014)."

simple typos: line 306: masses instead of messes
Response: It was corrected. Thanks for finding this.

line 440 and 441, in these lines the authors use a "," to indicate number greater than 1000, while in manuscript generally no comma is used. please adjust the numbers
Response: It was corrected.

line 270-271 : please adjust references: this format (Vehkamäki et al., 2004) or this format? Suni et al. (2008)
Response: It was corrected.

several lines: when indicating multiple references, please add a space after ","
Response: It was corrected.

about figure: I suggest to add the aerosol origin classification, as showed in figure 3, also in figure 2.

Response: Figure 2 was improved as suggested by the reviewer.

in figure 7 the averaged size distribution are showed, starting from more than 7 nm to 300 nm. Could the author describe how the 2 smps dataset (nano and standard) are combined? The nano measures from 3 nm to 80 nm, while the standard collect data from 10 nm to 300 nm.

To obtain the number size distribution in the size range from 7 nm to 300 nm as shown in Figure 7, we used nano SMPS data from 7 nm to 64 nm and standard SMPS data from 64 nm to 300 nm. We confirmed that nano SMPS and standard SMPS data agreed within ~8% in their overlapping size range (10 – 64 nm) (Figure S4). In fact, Watson et al. (2011) compared four types of scanning mobility particle sizers (i.e., TSI nano SMPS, TSI standard SMPS, Grimm SMPS, MSP WPS). They also demonstrated that hourly average particle concentrations in the 10 – 84 nm size range measured with TSI nano SMPS were well correlated with those measured with TSI standard SMPS. To clarify this issue, Figure S4 and description were added as given below.

Page 13, Line 334: "To obtain the number size distribution in the size range from 7 nm to 300 nm as shown in Fig. 7, we used nano SMPS data from 7 nm to 64 nm and standard SMPS data from 64 nm to 300 nm. The nano SMPS and standard SMPS data agreed within ~8.8% in their overlapping size range (10 – 64 nm) (Fig. S4), similar to a previous study (Watson et al., 2011)."

**Newly added references**

Hinds, W. C.: Aerosol Technology: Properties, Behavior, and Measurement of Airborne Particles, 2nd edn., Wiley-Interscience, New York, 1999.

Collins, D. B., Burkart, J., Chang, R. Y.-W., Lizotte, M., Boivin-Rioux, A., Blais, M., Mungall, E. L., Boyer, M., Irish, V. E., Massé, G., Kunkel, D., Tremblay, J.-É., Papakyriakou, T., Bertram, A. K., Bozem, H., Gosselin, M., Levasseur, M., and Abbatt, J. P. D.: Frequent ultrafine particle formation and growth in Canadian Arctic marine and coastal environments, Atmos. Chem. Phys., 17, 13119–13138, https://doi.org/10.5194/acp-17-13119-2017, 2017.

Dong, X., Xi, B., Crosby, K., Long, C. N., Stone, R. S., and Shupe, M.: A 10 year climatology of Arctic cloud fraction and radiative forcing at Barrow, Alaska, J Geophys Res-Atmos, 115, D17212, doi.org/10.1029/2009JD013489, 2010.

Watson, J. G., Chow, J. C., Sodeman, D. A., Lowenthal, D. H., Chang, M. C. O., Park, K., and Wang, X.: Comparison of four scanning mobility particle sizers at the Fresno Supersite, Particuology, 9, 204-209, 2011.

This manuscript describes approximately one month of measurements of aerosol physical properties and CCN concentrations taken aboard ship in the Pacific region of the Arctic, and Northern Pacific Ocean. This work demonstrates that terrestrial inputs in coastal areas can be an important driver of the particle size distribution in Arctic regions, and can drive strong new particle formation relative to open ocean areas of the Pacific. This is an important and interesting conclusion that has not been made in the past. Also, notably, this work presents measurements of particle size distributions starting at 3nm, which has not been done in many Arctic regions. This manuscript is well written, and presents the main results clearly. I support publication in ACP once the following issues can be addressed.

We sincerely appreciate all valuable comments and suggestions, which helped us to greatly improve the quality of the manuscript. We corrected the manuscript point by point accordingly.

**Major comments:**

(1) I am somewhat surprised that the authors have not demonstrated agreement between the two SMPS systems in their overlapping size range (i.e., 10-80nm). This is a crucially important size range for understanding particle growth and adequate quality control of these measurements should be addressed. How does N10-80 measured by the two instrument compare across the measurement period? Were any size calibrations of the two instruments made in the field? A clearer description of how the two measurements of particle size were aggregated over the overlapping range to produce Figure 7 is needed.

To obtain the number size distribution in the size range from 7 nm to 300 nm as shown in Figure 7, we used nano SMPS data from 7 nm to 64 nm and standard SMPS data from 64 nm to 300 nm. We confirmed that nano SMPS and standard SMPS data agreed within ~8% in their overlapping size range (10 – 64 nm) (Figure S4). In fact, Watson et al. (2011) compared four types of scanning mobility particle sizers (i.e., TSI nano SMPS, TSI standard SMPS, Grimm SMPS, MSP WPS). They also demonstrated that hourly average particle concentrations in the 10 – 84 nm size range measured with TSI nano SMPS were well correlated with those measured with TSI standard SMPS. To clarify this issue, Figure S4 and description were added as given below.

Page 13, Line 334: "To obtain the number size distribution in the size range from 7 nm to 300 nm as shown in Fig. 7, we used nano SMPS data from 7 nm to 64 nm and standard SMPS data from 64 nm to 300 nm. The nano SMPS and standard SMPS data agreed within ~8.8% in their overlapping size range (10 – 64 nm) (Fig. S4), similar to a previous study (Watson et al., 2011)."

(2) How does the condensation sink and average growth rate compare (1) across the different types of air masses sampled, and (2) to previous Arctic measurements? Was the condensation sink significantly higher in the Pacific air masses? Figure 7 would suggest that is the case. I

suggest that the authors make full use of their unique data set by calculating the above two quantities wherever possible, and including them in relevant figures. Statistics of these quantities could be included in Table 1. This would facilitate improved comparison with other studies in similar coastal Arctic regions (e.g., Collins et al., ACP, 2017 https://doi.org/10.5194/acp-17-13119-2017 and Burkart et al., GRL, 2017 doi: 10.1002/2017gl075671). In particular, Collins et al 2017 provides a detailed comparison of the condensation sink in their observations with that sampled globally (their Figure 8). Placing the author's results in this broader context would strengthen the paper significantly.

Response: We calculated growth rate and condensation sink for Arctic marine, Arctic terrestrial and Pacific marine air masses. The resulting values are given in Table 1 and compared with other studies in Arctic regions as given below.

Page 17, Line 422: "Table 1 shows particle growth rate (GR) and condensation sink (CS) for Arctic marine, Arctic terrestrial and Pacific marine air masses. The GR was calculated by fitting a linear regression to the peak diameter of the aerosol size distribution for the nucleation-mode between 4 and 20 nm against time during the NPF cases (Dal Maso et al., 2005; Pierce et al., 2014). The GR observed during the Arctic marine and Arctic terrestrial air masses were the 0.4 ± 0.3 nm h$^{-1}$ and 0.8 ± 0.2 nm h$^{-1}$, respectively, which was similar to the values previously observed from other Arctic regions. A shipboard expedition conducted during the summers of 2014 and 2016 throughout the Canadian Arctic, indicated that the GR varied widely from 0.2 to 15.3 nm h$^{-1}$ (Collins et al., 2017). The GR observed at Summit, Greenland was 0.2 ± 0.1 nm h$^{-1}$ (range of 0.09 to 0.3 nm h$^{-1}$) (Ziemba et al., 2010). Similarly, in Utqiagivik (Barrow), Alaska, the GR was 1.0 nm h$^{-1}$ in air mass influenced by Beaufort Sea, whereas the value was 11.1 nm h$^{-1}$ in air mass influenced by Prudhoe Bay (i.e., oil field area) (Kolesar et al., 2017)."

Page 17, Line 439: "The CS is a key parameter assessing the NPF and growth and determines how rapidly gaseous molecules condense onto pre-existing particles. The CS was calculated, following Dal Maso et al. (2002) and Collines et al. (2017). The resulting CS values are given in Table 1. The CS observed during the Arctic marine and Arctic terrestrial air masses were 0.5 ± 0.4 nm h$^{-1}$ and 0.9 ± 0.5 nm h$^{-1}$, respectively. The CS in this study is on the low end of the values observed during the summer in Arctic marine boundary layer (shipborne expeditions) (Collins et al., 2017), Utqiagivik, Alaska (Kolesar et al., 2017), and Ny-Ålesund, Svalbard (Giamarelou et al., 2016). In case when air mass passed over the Pacific Ocean, the CS was 2 or 4 times higher than those of Arctic air masses. It seems that such higher CS for Pacific marine air masses lowered the concentration of condensable vapors, thereby resulting in the non-event days in Pacific marine air masses."

(3) What other environmental variables were different between the three air mass source regions? e.g., wind speeds, cloud cover?

Response: Following the reviewer's comments, environmental variables such as wind speeds and wind direction for the three air masses were also shown in Table 1.

(4) It appears as though multiple modes were present in some cases (e.g., during 9/13-9/17), and based on Figure 7 appear in average size distributions. As discussed in Burkart et al., GRL, 2017 (doi: 10.1002/2017gl075671), comparing the growth rates of different particles modes can provide significant insight into the physical chemical properties of the condensing/partitioning species. I do not suggest that the authors undertake a detailed modelling exercise as in Burkart et al, rather that the authors calculate growth rates for the different modes and discuss what these quantities might mean in the context of the Burkart et

al analysis.

Response: To clarify this issue, we determined the growth rate of the distinct modes based on Burkart et al analysis. Figure S6 and a few sentences were newly added as follows.

Page 17, Line 432: "Particularly, simultaneous growth of multiple modes was present in some cases (9/13/2017− 9/21/2017). We calculated the GR of the distinct modes, as shown in Fig. S6. The results showed that growth of the larger mode (e.g., preexisting mode) was faster than the smaller mode (e.g., nucleation mode), consistent with ship-based aerosol measurements in the summertime Arctic by Burkart et al. (2017b). They proposed that growth was largely via condensation of semi-volatile organic material, because lower volatile organics could lead to faster growth of the smaller mode."

**Minor comments:**

L67-68: I agree with this statement in general; however, the authors miss two very relevant studies of ship-based coastal Arctic measurements: Collins et al., ACP, 2017 https://doi.org/10.5194/acp-17-13119-2017 and Burkart et al., GRL, 2017 doi: 10.1002/2017gl075671
Response: As suggested by the reviewer, we added the two references (Collins et al., 2017 and Burkart et. al., 2017).

L69-82: This paragraph focuses on studies that have linked NPF events to increases in measured CCN; however, the authors focus not only on studies from Arctic regions and seem to focus on global observations as the expense of being exhaustive in discussing all of the very few relevant Arctic studies. The two studies mentioned in the comment above should be included here, and are more relevant to the authors' discussion than Kalivitis 2015 and Rose 2017. Burkart et al., ACP, 2017 (doi: 10.5194/acp-17-5515-2017) also connect NPF with CCN in coastal Arctic environments.
Response: These references (Collins et al., 2017 and Burkart et. al., 2017) were addressed in the manuscript (Line 67, Line 70, Line 75 and Line 77), and the sentence was modified as given below.

Page 3, Line 75: "Several examples of increase in the CCN concentration after a few hours from the beginning of NPF events were presented by Burkart et al. (2017b) in the summer marine Arctic during the 2014 NETCARE Amundsen ice breaker campaign, by Kim et al (2019) at King Sejong Station in the Antarctic Peninsula, by Pierce et al. (2012) in a forested mountain valley in western Canada, and by Willis et al. (2016) in an Arctic aircraft campaign in Nunavut, Canada."

L123-126: Time resolution of the aethelometer?
Response: Raw time resolution of the aethelometer is 5 min. We added this information in the experimental section.

Page 5, Line 123: "The BC concentration was measured using an aethalometer (AE22, Magee Scientific Co., USA) with a 5-min time resolution to assess the influence of anthropogenic sources (such as local pollution and ship emissions)."

L141: Can the authors be more quantitative in describing "particularly high (spike) and varied dramatically in a short time"?

Response: The statement was modified to make quantitative description:

Page 6, Line 143: "The data collected when total aerosol number concentrations were higher than 8000 cm$^{-3}$ were removed. In addition, the CPC and SMPS data were removed for the time periods when total aerosol number concentrations suddenly increased more than two times higher than the background values."

L235: Tremblay et al, ACP, 2019 (https://doi.org/10.5194/acp-19-5589-2019) demonstrated regional growth events taking place over the Northern Canadian Arctic Archipelago, and their results are relevant to this discussion.
Response: We agreed with the reviewer's comments. We added the reference in the manuscript (Page 10 and Line 240) and the description was rewritten as given below.

Page 10, Line 243: "Tremblay et al (2019) also concluded that particle nucleation events occurred over spatial scales of at least 500 km during the summertime in the Canadian High Arctic."

L262-263: Could a satellite-based measurement of cloud fraction (e.g., from Aqua MODIS that is openly available) be used to assess any impact of cloudiness on the observations of NPF?
Response: To classify this issue, we compared weekly averaged liquid cloud fraction from Aqua/MODIS retrieval data for each period (week 1: 8/29/2017– 9/5/2017, week 2: 9/6/2017– 9/13/2017, week 3: 9/14/2017– 9/21/2017, and week 4: week 4 (9/22/2017– 9/29/2017)). Then, Figure S5 was added and the sentence (including references) was rewritten as given below.

Page 11, Line 270: "In addition, cloudiness which usually affects the real solar radiation reaching the surface was compared based on Moderate Resolution Imaging Spectroradiometer (MODIS) cloud fraction retrievals (Fig. S5). The data showed that cloud fraction was significantly high during the entire sampling periods, in general agreement with some other studies over the western Arctic region (e.g., Dong et al., 2010; Collines et al., 2017). In detail, the cloud fraction was relatively low for week 1 (8/29/2017– 9/5/2017; Fig. S5a) and week 3 (9/14/2017– 9/21/2017; Fig. S5c) when NPF event and growth was frequently observed (Fig. 3). This suggests that solar radiation at the surface, which is affected both by the cloud cover and SZA, may have influenced aerosol concentration and NPF observed here."

L422-435: Burkart et al., ACP, 2017 (doi: 10.5194/acp-17-5515-2017) also estimated the CCN activation diameter to be approximately 80nm in a coastal Arctic environment.
Response: We added the reference in the manuscript as given below.

Page 19, Line 478: "For instance, the $D_c$ of 80 nm at 0.6 % SS was observed during the aircraft measurement in July 2014 in the high Arctic marine boundary layer of Resolute Bay, Nunavut, Canada (Burkart et al., 2017a)."

Table 1: Do $N_{StandardSMPS}$ and $N_{nanoSMPS}$ include the overlapping size range? I suggest that the authors be explicit about the size range here, rather than referring to the instrument. Presenting mean concentrations in specific size ranges from aggregated size distribution data may be more useful (e.g., as shown in figure 2)
Response: As suggested by the reviewer, we presented mean concentrations of nucleation mode (3 − 20 nm) ($N_{NUC}$), Aitken mode (20 − 100 nm) ($N_{AIT}$), accumulation mode (100 − 300 nm) ($N_{ACC}$), and coarse mode (> 300 nm from OPS) ($N_{OPS}$) particles for the three selected periods

in Table 1.

Figure 7: Specify which SMPS is used in the size range from 10-80nm
Response: This is similar to a comment above. Again, we used nano SMPS data from 7 nm to 64 nm and standard SMPS data from 64 nm to 300 nm. We confirmed that the nano SMPS and standard SMPS data agreed within ~8.8% in their overlapping size range (10 – 64 nm), as shown in Fig S4.

**Specific comments:**
L61: Ice and snow-covered regions
Response: It was corrected.

L264: change 'trends' to 'tends'
Response: It was corrected.

L286: missing 'were'
Response: It was corrected.

**Newly added references**

Burkart, J., Hodshire, A. L., Mungall, E. L., Pierce, J. R., Collins, D. B., Ladino, L. A., Lee, A. K. Y., Irish, V., Wentzell, J. J. B., Liggio, J., Papakyriakou, T., Murphy, J., Abbatt, J.: Organic condensation and particle growth to CCN sizes in the summertimemarine Arctic is driven by materials more semivolatile than at continental sites, Geophys. Res. Lett. 44, 10725–10734. https://doi.org/10.1002/2017GL075671, 2017.

Burkart, J., Willis, M. D., Bozem, H., Thomas, J. L., Law, K., Hoor, P., Aliabadi, A. A., Köllner, F., Schneider, J., Herber, A., Abbatt, J. P. D., and Leaitch, W. R.: Summertime observations of elevated levels of ultrafine particles in the high Arctic marine boundary layer, Atmos. Chem. Phys., 17, 5515-5535, 10.5194/acp-17-5515-2017, 2017.

Collins, D. B., Burkart, J., Chang, R. Y.-W., Lizotte, M., Boivin-Rioux, A., Blais, M., Mungall, E. L., Boyer, M., Irish, V. E., Massé, G., Kunkel, D., Tremblay, J.-É., Papakyriakou, T., Bertram, A. K., Bozem, H., Gosselin, M., Levasseur, M., and Abbatt, J. P. D.: Frequent ultrafine particle formation and growth in Canadian Arctic marine and coastal environments, Atmos. Chem. Phys., 17, 13119–13138, https://doi.org/10.5194/acp-17-13119-2017, 2017.

Dal Maso, M.: Condensation and coagulation sinks and formation of nucleation mode particles in coastal and boreal forest boundary layers, J. Geophys. Res., 107, 10.1029/2001jd001053, 2002.

Dal Maso, M., Kulmala, M., Riipinen, I., Wagner, R., Hussein, T., Aalto, P. P., and Lehtinen, K. E. J.: Formation and growth of fresh atmospheric aerosols: Eight years of aerosol size distribution data from SMEAR II, Hyytiälä, Finland, Boreal Environment Research, 10, 323-336, 2005.

Dong, X., Xi, B., Crosby, K., Long, C. N., Stone, R. S., and Shupe, M.: A 10 year climatology of Arctic cloud fraction and radiative forcing at Barrow, Alaska, J Geophys Res-Atmos, 115, D17212, doi.org/10.1029/2009JD013489, 2010.

Giamarelou, M., Eleftheriadis, K., Nyeki, S., Tunved, P., Torseth, K., and Biskos, G.: Indirect evidence of the composition of nucleation mode atmospheric particles in the high Arctic, J. Geophys. Res.-Atmos., 121, 965–975, https://doi.org/10.1002/2015JD023646, 2016.

Pierce, J. R., Westervelt, D. M., Atwood, S. A., Barnes, E. A., and Leaitch, W. R.: New-particle

formation, growth and climate-relevant particle production in Egbert, Canada: analysis from 1 year of size-distribution observations, Atmos. Chem. Phys., 14, 8647-8663, 10.5194/acp-14-8647-2014, 2014.

Tremblay, S., Picard, J.-C., Bachelder, J. O., Lutsch, E., Strong, K., Fogal, P., Leaitch, W. R., Sharma, S., Kolonjari, F., Cox, C. J., Chang, R. Y.-W., and Hayes, P. L.: Characterization of aerosol growth events over Ellesmere Island during the summers of 2015 and 2016, Atmos. Chem. Phys., 19, 5589–5604, https://doi.org/10.5194/acp-19-5589-2019, 2019.

Watson, J. G., Chow, J. C., Sodeman, D. A., Lowenthal, D. H., Chang, M. C. O., Park, K., and Wang, X.: Comparison of four scanning mobility particle sizers at the Fresno Supersite, Particuology, 9, 204-209, 2011.

Ziemba, L. D., Dibb, J. E., Griffin, R. J., Huey, L. G., and Beckman, P.: Observations of particle growth at a remote, Arctic site, Atmos. Environ., 44, 1649–1657, https://doi.org/10.1016/j.atmosenv.2010.01.032, 2010.

---

## Author Response (AR2)

**Editor Decision: Publish subject to technical corrections (01 Apr 2020) by Manuela van Pinxteren**

The comments are addressed satisfactory and the manuscript can be accepted. However, I have one technical comments about the availability of the data for the public. Copernicus requests that all data should be made available on reliable (public) data repositories to assure open and reproducible science. There are numerous accessible databases available. If the data are not publicly accessible, a detailed explanation of why this is the case is required. Please address this comment before final publication.

Response: As suggested, all data (including the 1-hour average BC (KOPRI-KPDC-00001140), $CN_{10}$ and $CN_{2.5}$ (KOPRI-KPDC-00001137), standard SMPS and nano SMPS (KOPRI-KPDC-00001136), OPS (KOPRI-KPDC-00001138), and CCN (KOPRI-KPDC-00001141)) were uploaded to the Korea Polar Data Center (KPDC) ('https://kpdc.kopri.re.kr/search/'). To clarify this issue, we mentioned the data availability in the revised manuscript as given below.

Page 20, Line 519: "The data analyzed in this publication are available from the Korea Polar Data Center (KPDC) ('https://kpdc.kopri.re.kr/search/'), and also the raw data will be readily provided upon request to the corresponding author (yjyoon@kopri.re.kr)."